# The Selenoprotein Glutathione Peroxidase 4: From Molecular Mechanisms to Novel Therapeutic Opportunities

**DOI:** 10.3390/biomedicines10040891

**Published:** 2022-04-13

**Authors:** Kamari Weaver, Rachid Skouta

**Affiliations:** Department of Biology, University of Massachusetts, Amherst, MA 01003, USA; kweaver@umass.edu

**Keywords:** selenoprotein glutathione peroxidase 4 (GPX4), reduced glutathione (GSH), ferroptosis, lipid peroxidation, ferroptosis modulators, small molecules targeting GPX4

## Abstract

The selenoprotein glutathione peroxidase 4 (GPX4) is one of the main antioxidant mediators in the human body. Its central function involves the reduction of complex hydroperoxides into their respective alcohols often using reduced Glutathione (GSH) as a reducing agent. GPX4 has become a hotspot therapeutic target in biomedical research following its characterization as a chief regulator of ferroptosis, and its subsequent recognition as a specific pharmacological target for the treatment of an extensive variety of human diseases including cancers and neurodegenerative disorders. Several recent studies have provided insights into how GPX4 is distinguished from the rest of the glutathione peroxidase family, the unique biochemical properties of GPX4, how GPX4 is related to lipid peroxidation and ferroptosis, and how the enzyme may be modulated as a potential therapeutic target. This current report aims to review the literature underlying all these insights and present an up-to-date perspective on the current understanding of GPX4 as a potential therapeutic target.

## 1. Introduction

In this review, we summarize recent advances in our understanding of the phospholipid repair enzyme GPX4. We examine the molecular mechanisms of GPX4-mediated oxidative regulation, and its potential for therapeutic applications in human diseases. To capture the growing landscape of GPX4 biology and its impact as a therapeutic target, where appropriate, we refer readers to recent reviews for a more detailed discussion. 

## 2. Glutathione Peroxidase Family

### 2.1. The Origins of Glutathione Peroxidase

Glutathione peroxidases (GPXs) constitute an interesting family of phylogenetically related oxidoreductases observed in all living organisms. This family is a central component of the cellular antioxidant defense system. GPX activity was first detected in 1952, when G.C Mills and colleagues observed its activity in protecting hemoglobin from oxidative breakdown [1]. In the 1960s, GPX activity was noticed in other tissues, including the lungs and kidneys, and shown to reduce hydroperoxide-type compounds including H_2_O_2_ [2,3]. In the 1970s, the biochemical properties were characterized for GPX, and importantly, elemental selenium was discovered to play a key role in enzymatic activity [4,5,6,7]. This enzyme is known today as GPX1, and it is recognized as the most efficient peroxide removal system in the cytosol of many mammalian cells [8]. GPx-1 was the only GPx known until the 1980s. The list of mammalian GPxs has since grown up to eight, numbered one-through-eight. An intensive study on the evolutionary history of the GPx family was recently reviewed [9]. The GPX family is well established today for its redox properties in balancing oxidative homeostasis in cells. More specifically, GPX4 eliminates reactive oxygen species such as lipid peroxide to its benign alcohol analog using glutathione as a cofactor [10,11]. Moreover, GPx enzymes are also involved in the posttranslational modification of proteins, signaling pathways, among several other biological processes [12]. Table 1 and Table 2 depict the known range of the biological relevance of all eight GPX enzymes. 

### 2.2. The Role of Elemental Selenium in Selenoproteins 

The most unique feature of the mammalian GPX family is that many of them, apart from GPX-5, GPX-7, GPX-8, and GPX-6 in rodents, are selenoproteins [13]. Selenoproteins are rare proteins with strong antioxidant activity containing the 21st amino acid, selenocysteine. Selenocysteine (Sec), differs from cysteine (Cys) by a single atom, selenium replacing sulfur (Figure 1). Selenium incorporation into Sec is a highly complex process, owing to the complex genetic machinery required for both the synthesis of Sec-specific tRNA, and the co-translational use of the codon “UGA”, which also functions as the stop codon [31,32]. The biosynthesis of Sec is significantly more complicated than the other proteinogenic amino acids, including Cys [33]. Although Se incorporation is challenging, Se is essential for life and required for the proper enzymatic function of selenoproteins such as glutathione peroxidases (GPX) and thioredoxin reductases (TrxR) [34]. This raises an important question—what is the advantage of opting to use selenium when sulfur is structurally similar and requires fewer complex mechanisms to be utilized? This answer is not clear, although it may be explained by the difference in kinetic and thermodynamic properties of selenols compared to thiols [35,36]. A detailed chemical comparison between sulfur and selenium was recently examined, highlighting that the greatest difference between the two atoms occurs in their redox properties [37]. Se exhibits a dual effect, as nucleophile and electrophile, and showed better redox properties than sulfur. 

For example, Maiorino et al. reported that the substitution of sulfur with Se in enzymes bearing cysteine amino acid moiety leads to a substantially enhanced catalytic activity [38]. Accordingly, a recent study by Ingold et al. suggested that thiol-based catalysis is prone to overoxidation, whereas selenol-based catalysis is resistant to overoxidation [39]. Overoxidation of the selenoprotein would inactivate that protein, and thus prevent it from carrying out its oxidative functions. The superior redox properties of Se are a likely explanation for why Se is essential to selenoproteins despite its incorporation challenges.

In 1817, Jons Jacob Berzelius found, for the first time, a trace element of Se. Currently, Se is known as one of the most used micronutrients for humans. Se exhibits various health benefits such as antioxidant, anti-inflammatory, and anti-carcinogenic effects that are linked to its incorporation in selenoproteins [40,41]. The role of Se as a potential therapeutic target for cancer was recently reviewed [42]. Humans consume selenium through vegetables, meats, and dietary supplements. Se deficiency has been linked with adverse health conditions such as cardiovascular disease, infertility, myodegenerative diseases, and cognitive decline [43,44]. Se levels are highest in the kidney, followed by the liver, spleen, pancreas, heart, and brain. When Se uptake is deficient, these ranking changes depending on the priority order of these different organs for access to Se [45]. Interestingly, the brain retains Se longer than the other organs, highlighting the potential importance of Se in the central nervous system [46]. Indeed, a recent study identified the role of selenoproteins in brain function and implicated Se as a potential target in Alzheimer’s disease [47].

The GPX family is the major selenoprotein in the human body, but others such as the thioredoxin reductase family and the selenoprotein P family also exist. While the role of selenoproteins as an antioxidant is well established and biologically relevant, recent studies have shown that selenoproteins also have diverse roles unrelated to maintaining oxidative homeostasis [40]. Likewise, Selenium containing GPX enzymes (SecGPXs) can have diverse functions in the human body that extend beyond their principal role as an oxidoreductase. There are few human GPX enzymes that lack selenoprotein character and instead have a cysteine residue in their catalytic center rather than a selenocysteine residue. These enzymes are GPX5, GPX7, and GPX8. According to phylogeny studies, more than 700 Cys containing GPX-homologous sequences have been identified across many living organisms, indicating that only a small minority of GPXs are selenoproteins [8].

### 2.3. The Biochemical Structure of Glutathione Peroxidase 

A typical mammalian GPX possess a well-preserved catalytic tetrad consisting of glutamine, tryptophan, asparagine, and either a selenocysteine or cysteine residue [48]. (See Figure 2 for the amino acid sequence alignment comparing human GPX1-through-GPX8). The selenium residue which is positioned at a hydrogen-bonding distance from the remaining residues that make up the tetrad. 

The GPXs 1–3 and GPXs 5–6 enzymes are homotetramers.

On the other hand, GPX4, GPX7, and GPX8 enzymes are monomers. These monomeric enzymes possess unique electrostatic environment at the active site [49]. The unique monomer structure is speculated to allow the reduction of more complex lipid peroxides, although this has only been proven for GPX4. All GPX homologs display a tertiary structure conserved in thioredoxins and other oxidoreductase families. This conserved structure contains four α-helices that are localized near the protein surface, and seven beta-strands, five of which cluster together forming a central b-sheet [8] (see Figure 3 for the structural representation of GPX4). 

### 2.4. General Mechanism of Glutathione Peroxidase 

All Glutathione Peroxidases (GPXs) display a ping-pong mechanism, a type of non-sequential mechanism [49]. The well-known GPX mechanism involves two steps—an oxidation reaction followed by a reduction reaction [13]. The first step consists of the oxidation of the reduced GPX enzyme hydroperoxide-type compound, while the second step is a series of reductions of the oxidized GPX enzyme by a thiol-containing compound such as the reduced cofactor glutathione (GSH). In the GPX catalytic site and during the initial step, the selenocysteine or cysteine gets oxidized to selenic or cysteinic acids. At the same time, the toxic hydroperoxide gets converted into its respective benign alcohol. This step is believed to occur without the standard formation of an enzyme-substrate complex, which allows an incredibly rapid reduction of the substrate. Indeed, the rate constant for the oxidation phase is near 10^8^ M^−1^ S^−1^, which is among the fastest ever determined for biomolecular enzymatic reactions [8]. The reduction phase involves two subsequent steps, the first producing a glutathionylated intermediate GPX enzyme where the oxidized GPX reacts with the first equivalent of GSH. In the next step, a second equivalent of GSH reacts with the glutathionylated intermediate to form the stable product GSSG, and this allows the release of the reduced selenium or cystine for the next catalytic cycle (see Figure 4 for an example of GPX4 oxidoreductase mechanism).

### 2.5. Substrate Specificity 

Glutathione peroxidases (GPXs) have a diverse range of oxidizing substrates. Most GPXs reduce small organic hydroperoxides but are unable to reduce complex hydroperoxides such as lipid hydroperoxides or cholesterol. The one exception to this is GPX4, which is unique in being the only GPX enzyme capable of reducing large and complex lipid hydroperoxides and cholesterols, even when they are embedded in the biological membrane [21]. Glutathione (GSH) is the most preferred co-factor reducing substrate in mammalian GPX enzymes, although GPX4 has the unique ability to utilize other protein thiols in addition to GSH [51].

### 2.6. Role of Oxidative Stress in Biology 

The GPX’s primary role as an oxidoreductase is significant as it helps to prevent oxidative stress and oxidative damage, which are prominent biomarkers for human disease [52]. GPX’s are well-known for their role in disorders characterized by oxidative injuries [1,8,13]. Recent studies showed that oxidative injuries were involved in neurodegeneration such as Alzheimer’s and Parkinson’s disease [53,54], cancer [55,56], diabetes [57], and cardiovascular diseases [58,59]. Oxidative stress is characterized by an imbalance between prooxidant species (e.g., reactive oxygen species (ROS) and reactive nitrogen species (RNS) that promote oxidation, and antioxidant enzymes (e.g., GPX and transferrin) that inhibit this oxidation. 

The imbalanced ROS in the body can cause damage to DNA, lipids, and protein, all of which would cause injury to the cell and can even lead to cell death. The main cause of oxidative stress is an increased production or accumulation of ROS and RNS, a known class of highly reactive compounds containing oxygen and nitrogen respectively. This ROS class of compounds contains radicals and non-radical species including superoxide hydroxyl radicals and hydrogen peroxide, respectively. ROS are mainly derived from the mitochondria where they are regularly produced as a natural byproduct of metabolism [55]. During energy metabolism, the mitochondria’s central role is in oxidative phosphorylation which generates ATP through processes including the TCA cycle and the electron transport chain (ETC). A recent paper made the discovery that inhibition of the mitochondrial TCA cycle and ETC minimized lipid peroxide accumulation and ferroptosis. Due to the large abundance of ROS in the mitochondria, antioxidants such as GPX4 plays a major role in protecting the mitochondria from oxidative damage. Indeed, GPX4 knockdown lead to increased mitochondrial ROS levels and a subsequent decrease in mitochondrial membrane potential. 

Interestingly, ROS play a dual role in the human body. An excess of ROS may cause the oxidation of macromolecules and organelles in the body, leading to oxidative stress.

However, ROS may also have non-deleterious effects. They often function as physiological regulators of intracellular signaling pathways, mediate the redox modifications of proteins, and function as intracellular messengers, all crucial roles in the human body. Since ROS is both important in human physiology and can elicit negative effects, the proper regulation of ROS/redox homeostasis is crucial. This maintenance of cellular homeostasis is achieved primarily through the glutathione peroxidase and other oxidoreductase families, among dietary antioxidants and tumor suppressors. Therefore, glutathione peroxidases play a crucial role in health and diseases.

### 2.7. Glutathione Peroxidase in Health and Diseases 

An intensive review comparing the history, biochemistry, genetics, and biomedical relevance of each GPX enzyme was recently published [13]. Briefly, the GPX 1–8 genes are mapped to chromosomes 3, 14, 5, 19, 6, 6, 1, and 5, respectively. While all GPXs are generally involved in the detoxification of prooxidants, each GPX has distinct roles in biology [13]. GPX1 is most known for maintaining peroxide homeostasis in the insulin signaling pathway [14], along with its complex role in cancer as an antiapoptotic enzyme [15]. GPX2 plays a unique role in carcinogenesis [16,17,18]. GPX3 acts as a tumor suppressor [19,20]. GPX4 is involved in the regulation of cell death including ferroptosis [22], and together with GPX5, in male fertility [23,27].The functions of GPX6 are less well known, but recently it has been implicated as modulator of Huntington’s disease [13,28]. GPX7 and GPX8 are speculated to be involved in protein folding [29,30] (See Table 2 for a comparison of the biological properties of each enzyme). Although the functions of GPX1—GPX8 extend much beyond their oxidative roles, GPX4 is the only GPX that when knocked out (GPX4KO) is lethal. This suggests that GPX4 exhibits specific biochemical and molecular functions that are essential for human health and diseases, thereby making the enzyme an attractive target for novel therapeutic opportunities. 

## 3. Glutathione Peroxidase 4 in Biochemistry and Molecular Biology 

### 3.1. The Origins of Glutathione Peroxidase 4

GPX4, initially called phospholipid hydroperoxide glutathione peroxidase (PHGPX), was first purified in 1982 by Ursini et al. [60]. Ursini and colleagues were studying the protein contents of the cell sap, a fluid found in the living cell of the small cavities of vacuoles, from pig liver when they found glutathione peroxidase that had a unique ability to protect liposomes and biological membranes from oxidative degradation. Indeed, the ability to reduce complex hydroperoxides including the phospholipids embedded in biological membranes remains a distinctive and crucial feature of this enzyme. PHGPX was later found to contain selenocysteine and to have a unique monomer structure [61]. In 1991, primary structure studies revealed that PHGPX was a new selenoprotein distinct but related to GPX-1, allowing PHGPX to be considered a new member of the glutathione peroxidase family [62]. Today, PDHGPX is known as GPX4 and is recognized as a key mediator of a variety of human diseases including rare genetic disorders [25]. Compared to the other GPXs family, GPX4 possess a broader substrate specificity, and play a vital role in early mouse development, and its involvement in an impressive variety of biological processes [63]. In addition to the small hydroperoxides that are common substrates for all GPX enzymes, only GPX4 can reduce complex phospholipid hydroperoxides specifically, even those embedded in the biological membrane, to their corresponding benign alcohols. This function is of particular therapeutic interest as it suggests GPX4′s role in maintaining the integrity of biological membranes. In addition, GPX4 has a broader substrate specificity than other GPX enzymes as it is not limited to only glutathione as a reductant. Rather, GPX4 has the unique ability to get reduced by thiol-containing proteins as well [64,65] which increases its biological relevance. 

### 3.2. Clinical Relevance of Glutathione Peroxidase 4 

GPX4 represents an intriguing specific target for new pharmacological treatments, precision therapies, and other therapeutics. The role of GPX4 as a primary cellular defensive system against oxidative damage, suggests the significance of GPX4 in disorders characterized by oxidative injury. Recently, GPX4 was discovered to be involved in a rare neonatal lethal disorder called Sedaghatian-type spondylometaphyseal dysplasia (SSMD). In several cases of SSMD identified to date, patients presented with a variant in GPX4. The R152H variant of GPX4, a missense mutation that leads to a significant loss in enzymatic activity, was observed in three patients with SSMD. A comprehensive investigation of this variant and potential therapeutic treatments was recently published [25]. The precise mechanism involving GPX4-induced SSMD remains to be discovered, although it is established that a complete loss of enzymatic GPX4 activity is a central component to the clinical phenotypes of SSMD patients. Importantly, the implication of GPX4 in SSMD has inspired an organization called CureGPX4. A detailed discussion of the CureGPX4 goal in facilitating the development of new therapies for rare disorders was reviewed recently [25]. In addition to genetic disorders, GPX4 has emerged as a hotspot in biomedical research following its recognition as a chief regulator of an emerging form of regulated cell death called ferroptosis [22]. Ferroptosis is characterized by the iron-dependent accumulation of toxic lipid peroxides that are normally maintained by antioxidants such as GPX4. A complete discussion of ferroptosis and its relevance to a multitude of diseases will be discussed later in this review.

GPX4 is also recognized as essential for proper embryo development, as GPX4 knockout mice die in utero [66,67]. The therapeutic basis of GPX4 in cancer, [26,68], neurodegenerative disorders [69,70,71], and male infertility [23,24] are well-investigated. 

### 3.3. Structure and Genetics of Glutathione Peroxidase 4 

The human GPX4 gene is located on chromosome 19 at band 19p13.3. The gene contains three different splice variants: cytosolic (c-GPX4), mitochondrial (m-GPX4), and nuclear (n-GPX4) [8]. According to the Uniport data bank, GPX4 contains 197 residues with a molecular mass of 22 kDa. GPX4 presents with a typical thioredoxin motif containing four α-helices that are localized near the protein surface, and seven b-strands, five of which cluster together forming a central b-sheet (see Figure 3 for a crystal structure representation of GPX4 adapted from the PDB (6ELW)) [50].

GPX4 shares the conserved catalytically active tetrad observed in other SecGPXs containing selenocysteine (Sec46), glutamine (Gln81), tryptophan (Trp136) and asparagine (Asn137). Gln81 and Trp136 function to stabilize Sec46, especially the selenium redox intermediates (Se-, SeO-) through a network of hydrogen bonds. Asn137′s primary function is in the catalytic mechanism [72]. Mutations of any of these residues reduces GPX4 activity. Mutations of the selenocysteine to cysteine reduces GPX4 activity by 90%, highlighting the importance of the selenoprotein characteristic [21]. The monomeric GPX4 structure is missing an internal stretch of 20 amino acids that are conserved in the tetrameric GPX enzymes. These 20 residues code for a solvent-exposed loop surrounding the active site, and this loop may prevent complex substrates from accessing the active site. The lack of this loop in GPX4 is speculated to give GPX4 its unique ability to reduce large and complex substrates such as phospholipid peroxide. Crystal structures of cGPX4 have traditionally been solved using Sec46Cys mutants, rather than the native protein, due to the inherent challenge of expressing selenoprotein-containing proteins in sufficient quantities [73]. Human GPX4 Sec46Cys crystal structures are well-studied [74] and very similar to crystal and solution studies of mouse Sec46Cys mutants that have recently been solved [48]. The solving of a true, selenium-containing wild-type human protein would improve the understanding of GPX4 functional interactions and may facilitate the pharmacological development of GPX4-targeting compounds. The first crystallization and structural determination of a true wild-type SecGPX4 protein was solved recently by Moosmayer et al. [75]. This feat was accomplished using a novel approach first described in [75]. The wildtype selenium-containing crystal structure is structurally similar to previously reported mutant GPX4 structures as compared in a previous study [50]. The biggest structural difference between the Sec46Cys and Sec46 GPX4 variants was at the active site. For the Sec-containing enzyme, the hydrogen bond distance between the Sec46 and the Gln81 of the catalytic site is much shorter, at 3.2 Å. In a recent study by Labreque et al, the first NMR assignments of GPX4 were presented [49]. Together, this work provides a starting point for improving the understanding of the structural drivers of GPX4 interactions with its targets.

### 3.4. Enzymology and Kinetics of Glutathione Peroxidase 4 and GSH

Mechanistically, GPX4 displays catalytic oxidation/reduction steps involving the redox shuttling of the selenocysteine active site between an oxidized and a reduced state (Figure 4). GPX4 does not follow Michaelis–Menten kinetics, likely because the rate-limiting step during this mechanism is the binding of the initial peroxide to the GPX4 active site, rather than the elimination of the GPX-substrate complex. 

During the first phase, the oxidation of the active site selenol (Se-H) to selenenic acid (Se-OH) pf the GPX4 leads to the reduction of the toxic lipid peroxides to generate the non-toxic lipid alcohols (Figure 4). During the second phase, a reducing substrate such as GSH is used to reduce the selenenic acid back to the active selenol and close its catalytic cycle and allow the oxidation/reduction process to be repeated. The catalytic process is executed in two steps, starting with the reaction of GSH forming a selenium–glutathione intermediate. Next, the selenenic acid is reduced back to the selenol together with the release of glutathione disulfide (GS-SG) (Figure 4). Interestingly, a recent crystal structure showed the presence of a seleninic acid in the GPX4 active site [50]. Therefore, the suggested mechanistic model may consist of two consecutive phases: (i) the transferring of selenium between selenol (R-Se-H) and selenenic acid (R-SeO-H) followed by (ii) the transferring of selenenic acid (R-SeO-H) to seleninic acid (R-SeOO-H) although this has not been experimentally verified. It may be possible that depending on cellular conditions, GPX4 may utilize either the “low-oxidation” (R-SeO-) or the “high-oxidation” (R-SeOO) cycle. Central to the enzymatic mechanism of GPX4 is the thiol-containing tripeptide, Glutathione. Although GPX4 is unique from the other GPX enzymes in that it is not limited to GSH and may use other thiol-containing proteins as reductants as well. However, GSH is the most abundant in cells. In addition, the reducing form of GSH is implicated as an antioxidant marker for several human diseases [76,77,78]. In recent years it has become recognized as a key regulator, alongside GPX4, in an emerging regulated cell death pathway called ferroptosis [79]. Ferroptosis is a recently reported non-apoptotic programmed cell death mediated by the GSH/GPX4 cellular pathway. The role of GSH as the most abundant antioxidants synthesized in cells is well studied, established by their role in removing reactive oxygen species [80]. Glutathione is made up of L-glutamate, cystine, and glycine residues. Its synthesis is catalyzed first by glutamate cystine ligase (GCL), which couples to ATP hydrolysis to form an amide bond between the carboxyl group of glutamate and the amino group of cysteine. Glutathione synthase then adds glycine to the dipeptide to create GSH, a step that is also coupled with ATP hydrolysis. Glutathione exists in the reduced form (GSH) or oxidized glutathione disulfide form (GGSG). GSH exhibits a strong electron-donating character, owing to its sulfhydryl group on the cysteinyl position. As GSH loses electrons to become oxidized, two GSH molecules dimerize through a disulfide bridge to form GSSG [78]. Importantly, this dimerization is reversible through a reduction reaction. GPX4 uses glutathione as a cofactor to reduce peroxides to their corresponding alcohols, which is important in blocking the formation of reactive oxygen species and subsequently preventing oxidative damage. The GPX4, GSH, and GSSG team is the main antioxidant system that protects living organisms from oxidative stress and oxidative damage.

### 3.5. Synthesis, Degradation, and Regulation of Glutathione Peroxidase 4 

The protein synthesis and degradation of GPX4 are tightly regulated by a network of complex mechanisms. The presence of selenium is important in the synthesis of an enzymatically active GPX4, especially in relation to its role as a ferroptosis-mediator [39]. The cellular mevalonate pathway produces isopentyl pyrophosphate (IPP) which that facilitate GPX4 protein synthesis. Heat shock protein family A member 5 (HSPA5) is a molecular chaperone which binds to GPX4 to prevent its degradation [81]. HSPA5 is especially interesting as its functioning may be implicated in cancer cell resistance to ferroptosis. Heat shock protein 90 (HSP90) mediates GPX4 degradation, which has an opposite effect compared to HSPA5, as it facilitates ferroptosis.

## 4. Glutathione Peroxidase 4 as a Chief Regulator of Ferroptosis

### 4.1. Overview of Ferroptosis

Regulated cell death (RCD) is a highly controlled modality involving tightly structured signaling cascades and effector mechanisms. Proper RCD function is necessary for the development and maintenance of tissue homeostasis and as a mechanism for eliminating damaged cells. RCD is particularly relevant in neurodegenerative diseases, cancers, and the development of multicellular organisms [82]. Apoptosis was the first and most well-known RCD mechanism characterized in 1972 by John Kerr et al. [83]. The development of therapeutics for cancers and other diseases characterized by abnormalities in RCD requires a comprehensive understanding of the multitude of cell death mechanisms and their subtypes [84]. Cell death mechanisms in mammalian cells have been traditionally classified as apoptotic, necroptotic, or autophagic. In recent years, however, several novel forms of cell death have been discovered and it is thought that many more may exist. A form of regulated cell death named “ferroptosis” has become a hotspot in biomedical research as it plays a critical regulatory role in an impressive variety of human diseases. Ferroptosis was recently implicated as a mechanism of neurodegeneration in Alzheimer’s Disease [71], and as a mechanism for neuroinflammation in Parkinson’s Disease [85,86]. Ferroptosis is seen as a novel therapeutic for several neurodegenerative disorders including Alzheimer’s, Parkinson’s, and Huntington’s disease [87,88]. Studies involving ferroptosis in neurodegenerative disorders are generally aiming to inhibit ferroptosis cell death; however, inducing ferroptosis is also a promising therapy for disorders such as cancers. Indeed, ferroptosis is implicated in a vast number of cancers including prostate [89] and lung cancers [90,91]. The therapeutic potential of ferroptosis in human disease is extensive and its implication in a wide range of disorders is accelerating. [59,92,93]. Early research on ferroptosis-characteristic cell death is dated back to the 1950s and 1960s, led by Harry Eagle who demonstrated that cysteine depletion reduced glutathione and led to cell death, whereas cysteine synthesis protected cell death by restoring glutathione [94,95,96]. Additionally, an antioxidant named a-tocopherol (a type of vitamin E) rescued cell death independent of glutathione [97]. A group led by Joseph Coyle established glutamate-induced cell death dependent on the inhibition of cystine transport, which was later named oxytosis in 2001, and considered today as a subtype of ferroptosis [98,99]. In 2012, the term “Ferroptosis’’ was named and characterized as an additional programmed cell death that is sensitive to iron and lipid peroxidation in 2012 [100]. The features of ferroptotic cell death, especially in comparison to other forms of cell death, were described comprehensively in a recent paper [101]. Briefly, at a morphological level ferroptosis-induced cells have reduced mitochondrial volume, increased bilayer membrane density, and reduction of mitochondrial cristae. Biochemically, cells present with depletion of intracellular glutathione and subsequent decreased activity of GPX4. This foreshadows the vital role GPX4 plays in the regulation of ferroptotic cell death. The current understanding of the genetic basis of ferroptosis is incomplete and may be disease-state specific [102,103]. Broadly, at the genetic level ferroptosis is likely to involve genetic changes in iron homeostasis and lipid peroxidation metabolism. 

### 4.2. Molecular Mechanisms of Ferroptosis 

Ferroptosis is driven by lipid peroxidation and is regulated at multiple levels. Generally, it is characterized by cells accumulating lipid peroxides and failing to utilize the internal defense systems that should eliminate these lipid peroxides. This causes peroxide accumulation to lethal levels which will damage the phospholipids that make up the cell membrane and ultimately cause cell death. There are three main regulatory levels of ferroptosis: (1) Xc/GSH/GPX4, (2) NAD(P)H/FSP1/CoQ10, and (3) GCH1/BH4/DHFR. The system Xc/GSH/GPX4 pathway was the first discovered and today is recognized as the centerpiece of the ferroptosis mechanism (Figure 5) [22,66]. 

A characteristic feature of ferroptosis is the accumulation of phospholipid hydroperoxides in the presence of catalytically active iron. This process is naturally inhibited by the system Xc/GSH/GPX4 regulatory pathway. The most upstream component of this pathway is a heterometric antiporter known as system Xc, which is widely distributed in biological membranes. The antiporter mediates the cystine uptake in cells that is essential for the biosynthesis of GSH [104]. GSH also functions as a cofactor for GPX4-mediated antioxidant defense [105]. System Xc is made up of two central components, solute carrier family 3 member 2 (SLC3A2) and solute carrier family 7 member 11 (SLC7A11). A number of tumor suppressor genes, including p53 [106] and BRCA-1associated 1 (BAP1) [107], suppress SLCA11 and overall system Xc activity. Disturbing system Xc (e.g., glutamate, erastin, sulfasalazine, and sorafenib), depleting GSH (e.g., buthionine sulfoximine) or inhibiting GPX4 (e.g., RSL3, ML162, Fin56). In addition to the Xc/GSH/GPX4 pathway, 2 additional pathways are recognized as major pathways in the ferroptosis system; the NAD(P)H/FSP1/CoQ10 and the GCH1/BH4/DHFR systems. The details of these pathways have been described comprehensively in a recent review [108]. An important, yet somewhat controversial feature of ferroptosis is the understanding of how iron is involved [93]. The current understanding based on studies from several groups is that hydroperoxide-generating enzymes called lipoxygenases (LOXs) can initiate ferroptosis by generating lipid peroxides. Then, liable iron (iron not bound to enzymes) propagates these lipid peroxides and leads to the chain reaction of lipid peroxidation [109,110]. It is also believed that other iron-dependent enzymes may contribute to lipid peroxidation under specific circumstances.

### 4.3. Regulation of Ferroptosis through GPX4

GPX4 was first recognized in 2014 as a key regulator of ferroptotic cell death by Yang et al. [22]. Among the three known GPX4 isoforms (i.e., mitochondrial, cytosolic, and nuclear), only the cytosolic isoform (c-GPX4) is required for ferroptosis prevention. The unique function of GPX4 to reduce complex hydroperoxides including phospholipid and cholesterol hydroperoxides, protects biological membranes from the lipid peroxidation chain reaction that otherwise would lead to ferroptosis cell death. GPX4 is seen today in biomedicine as a popular genetic and pharmaceutical target for therapies looking to induce or suppress ferroptosis cell death [111]. As mentioned prior, oxidation of carbon molecules in the mitochondria during cellular respiration is essential for generating energy for the cell in the form of ATP, but it also gives rise to reactive oxygen species. In addition to the mitochondria, ROS may also be generated from lipid metabolism enzymes such as lipoxygenases. The accumulation of ROS can lead to the oxidation of the phospholipids embedded in the membrane, leading to cell death through lipid peroxidation accumulation. GPX4 is as a central mediator of this process, because of its role in converting toxic lipid hydroperoxides (R-OOH) to benign lipid alcohols (R-OH) using reduced GSH as a cofactor. This specific role of GPX4 prevents the accumulation of ROS and rescues cell membranes from damage. Additionally, GPX4 is a selenoprotein, thus a cells sensitivity to ferroptosis is related to selenium availability. Indeed, in studies delivering selenium to cells or animals it was found to suppress ferroptosis likely because it improves the oxidative efficacy of GPX4. Conditions that decrease the GPX4 levels or activity have an immediate impact on cell survival and health. GPX4-null mice undergo embryonic lethality, whereas conditional GPX4-mutant mice result in ferroptosis, neurodegeneration [112], loss of antiviral immunity [113], infertility, and ischemia-reperfusion injury in the kidney and liver [114].

### 4.4. Modulation of GPX4 to Probe Ferroptosis

Small-molecule drug discovery is an exciting process pharmaceutical researchers use as a tool for interrogating and perturbing biological systems [115,116]. Small-molecule drugs are organic compounds that have a low molecular weight and could target regulatory proteins as a method for probing biologically relevant molecular pathways such as ferroptosis [117]. It is well established that GPX4 is an attractive specific target for new pharmacological therapeutics aiming at activating cell death in cancer or inhibiting cell death in degenerative diseases. Viswanathan, et al., made the striking observation recently that cancer cells in a therapy-induced, drug-resistance state display high levels of polyunsaturated lipids and an acquired dependency on GPX4 [118]. Accordingly, increasing evidence encourages the rational combination of GPX4 inhibitors and current chemotherapies as a dual therapy in a diverse range of cancer types [26,119]. 

Although there is significant therapeutic benefit to pharmaceutically targeting GPX4, the unique structural features of the protein make it challenging for the development of small-molecule inducers and inhibitors. GPX4 lacks a drug-like binding pocket, and it relies on the nucleophilic selenocysteine residue for proper enzymatic activity. These features may mean that covalent inhibitors are required for inhibition of cellular GPX4, which often have poor selectivity and pharmacokinetic properties. Ferroptosis modulators can be generally classified into three categories: (1) system xc inhibitors, (2) GPX4 inhibitors, and (3) compounds that inhibit GPX4 activity via GSH depletion. The first chemical inducer of ferroptosis, identified from a diverse chemical library, was a small molecule named Erastin [120]. Erastin functions to induce ferroptosis through the inhibition of system xc. This results in depletion of glutathione, which subsequently impairs the antioxidant-ability of GPX4, leading to accumulated lipid peroxidation and ferroptotic cell death [121]. Erastin also binds directly to mitochondrial voltage-dependent anion channels. Similar molecules (including sorafenib, glutamate, and sulfasalazine) mimic this indirect Compounds that directly target GPX4 and are independent of system xc, are also promising probes for therapeutics (Table 3). GPX4 impairment through the inhibition of system xc have been discovered and classified as class 1 inducers (Table 4). A recent study by Wei, et al. demonstrated that directly targeting GPX4 is more effective than targeting GPX4 indirectly through the distribution of glutathione [122]. In 2008, a novel small molecule inducer of ferroptosis was identified named RSL3 [123]. RSL3 contains a reactive chloroacetamide moiety that makes it a potent and irreversible GPX4 inhibitor, although its poor solubility and poor absorption, distribution, metabolism, and excretion (ADME) properties limit its effectiveness in in-vivo studies [22]. Several chloroacetamide-containing inhibitors of GPX4 have been identified to date. In 2016, a ferroptotic inducer termed FIN56 was identified as an inducer that works through a dual mechanism of depleting GPX4 protein and mevalonate-pathway-derived coenzyme Q10 (CoQ10). It has become evident that ferroptosis is a druggable pathway where ferroptosis inducers may be used to kill malignant cells, and ferroptosis inhibitors may be used to prevent cell and tissue loss in degenerative disorders. The full scope of pharmaceutical targets for ferroptosis and GPX4 have been summarized intensively in several recent reviews [117,119,124].

### 4.5. GPX4, Ferroptosis, and Mitochondria

The mitochondria are double-membrane organelles that play a vital role in energy metabolism. As discussed in greater detail in Section 2.6 of this review, the energy-producing cellular processes such as the mitochondrial tricarboxylic acid cycle (TCA) cycle and the electron transport chain (ETC) create ROS that have the potential to be toxic to cells if not regulated properly by antioxidant defense mechanisms such as GPX4 [125,126]. Due to this relationship with ROS, the mitochondria may also be thought of as a key component of cell death pathways including ferroptosis. The appearance of shrunken mitochondria together with an increased membrane density are distinctive hallmarks of ferroptosis recognized since the first discovery of ferroptosis [100]. A comprehensive analysis of the mitochondrial metabolic processes that may facilitate ferroptosis, in addition to the mitochondrial defense systems (most notably GPX4) was recently conducted [127]. In summary, the mitochondria may be related to ferroptosis exclusively due to its metabolic processes that facilitate ferroptosis. However, the mitochondria may not be required for ferroptosis as in mitochondria-depleted cells, toxic accumulation of lipid peroxides are sufficient in inducing ferroptosis [127]. Furthermore, the mitochondria play a complex role in ferroptosis, and the details remain unknown. The mitochondria may be involved in the cysteine-depletion induced aspect of ferroptosis, rather than the GPX4 inactivation-induced pathway of ferroptosis. This hypothesis is a result of the discovery that GPX4 inactivation-induced ferroptosis is caused mostly by nonmitochondrial lipid peroxidation [66].

## 5. Conclusions and Perspectives 

The selenoprotein glutathione peroxidase 4 is quickly emerging as an intriguing therapeutic target, owing to its essential role in protecting cellular membranes from oxidative damage. GPX4′s primary function is in preventing toxic lipid hydroperoxide accumulation by converting hydroperoxides into their respective non-toxic alcohol. Crucially, GPX4 is recognized as the chief regulator of ferroptosis one of the recently identified regulated cell death mechanism. Ferroptosis relevance span a multitude of disorders ranging from cancers to neurodegenerative disorders, thereby suggesting the role of GPX4 in these disorders as well. The body of research on GPX4 is large, and accelerating, as researchers continue to uncover how GPX4 is implicated in human diseases and how inactivation of GPX4 leads to lipid peroxidation-induced ferroptosis. Notably, very recent studies have investigated GPX4 as a target for precision therapy in a rare genetic disorder. This rare neonatal lethal disorder Sedaghatian-type spondylometaphyseal dysplasia (SSMD) has been linked to four GPX4 variants [25]. Although the mechanism of pathogenesis is unclear, it is thought that the clinical phenotypes of SSMD patients involve complete loss of GPX4 enzymatic function. There remains much to learn about the roles, regulation, and therapeutic intervention of GPX4. A challenge in the coming era of GPX4 research will be improving the structural understanding of the wild-type GPX4 protein. Due to the challenges of solving the crystal structure of a selenium-containing protein, there are limited wild-type crystal structures for human GPX4 available. The continued development of a true selenium-containing enzyme may facilitate the pharmaceutical development of GPX4 targeting compounds. In the absence of new therapeutic interventions, GPX4-mutants will contribute to the pathogenesis of a growing number of diverse human diseases. Increasing understanding of the role and regulation of GPX4 will hopefully encourage clinical advancement in cancer therapy, neurodegenerative disorders, and more.

## Figures and Tables

**Figure 1 biomedicines-10-00891-f001:**
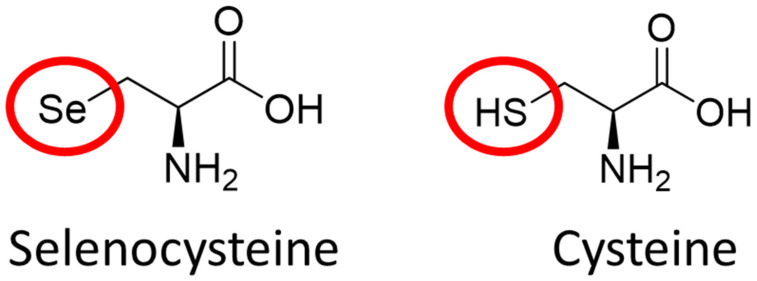
Chemical structure of selenocysteine and cysteine. The two compounds differ by a single atom, a selenium atom in selenocysteine and a sulfur atom in cysteine. The presence of selenium improves the redox properties of a given protein (e.g., GPX family, thioredoxin reductase, selenoprotein P).

**Figure 2 biomedicines-10-00891-f002:**
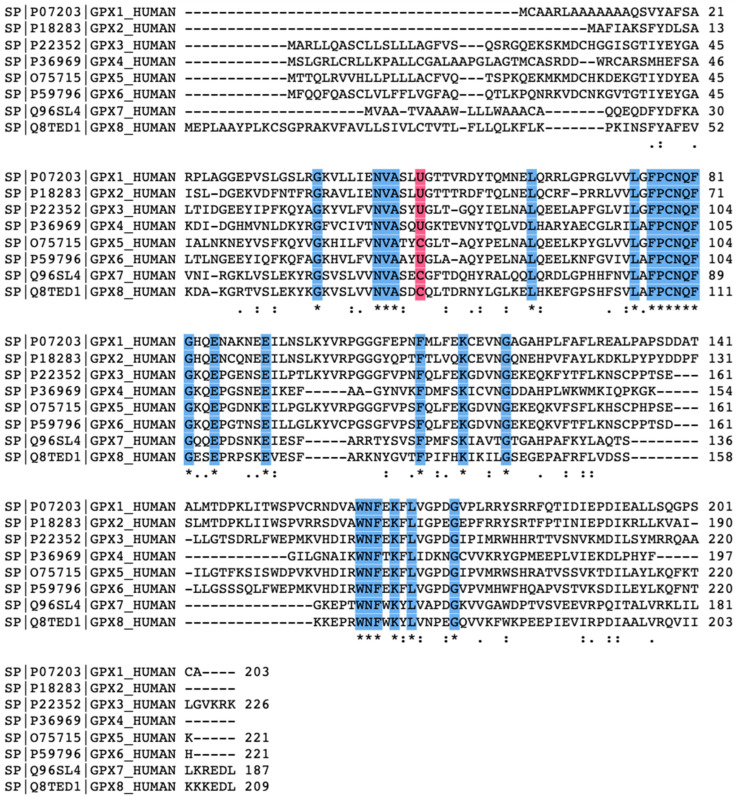
Amino acid sequence alignment adapted from Uniprot comparing human GPX1-through-GPX8. Blue highlighted regions fully conserved residues. The glutathione peroxidase family has overall low sequence similarity. The pink highlight recognizes the redox-active site (U: Selenocysteine; C: Cysteine). Symbols: An asterisk represents fully conserved residues. A colon represents conservation of residues with very similar chemical properties. A period represents conservation of resides with weakly similar properties.

**Figure 3 biomedicines-10-00891-f003:**
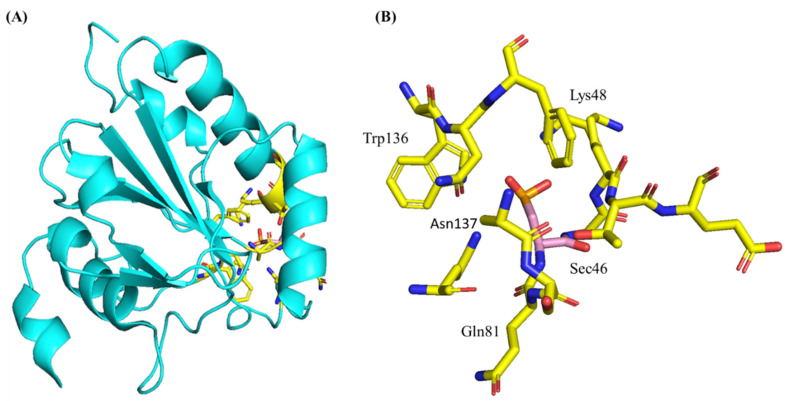
Crystal structure of the apo selenocysteine-containing human GPX4 protein adapted from the protein data bank (PBD code: 6ELW, 192 residues, 21.91 kDa). Structure was originally reported by Borchert et al. [50]. (**A**) Shows the full crystal structure of GPX4 (PDB entry 6ELW). (**B**) Shows the active site residues of the GPX4. Selenocysteine at position 46 in the active site (highlighted in pink color). Surrounding active site residues (highlighted in yellow color). The colors of atoms follow the default representation where dark blue represents a Nitrogen residue and red represents a Oxygen residue.

**Figure 4 biomedicines-10-00891-f004:**
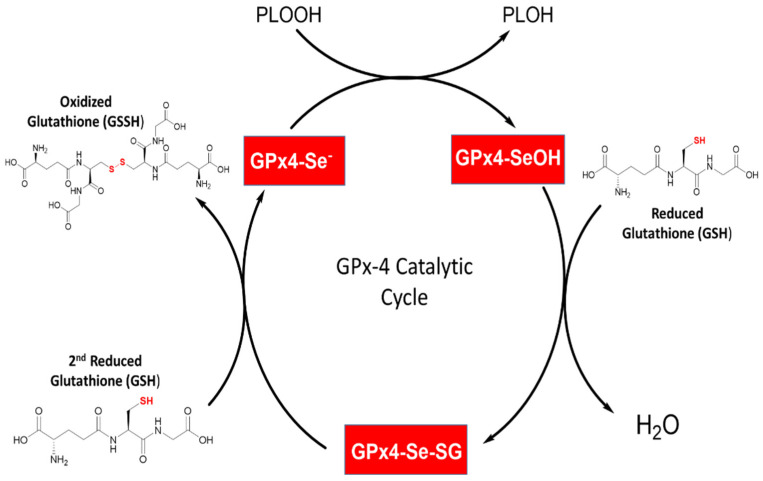
GPX4 catalytic cycle representation. GPX4 selenol (GPX4-SeH) gets oxidized into selenic acid (GPX4-SeOH). This oxidation powers the reduction of toxic lipid hydroperoxides into their respective alcohol. The selenic acid is reduced back to its active form selenol using two equivalents of GSH. The first equivalent of GSH reacts with selenic acid to form a selenium-glutathione intermediate and generate water. While the second GSH equivalent reacts to reduce the selenium-glutathione intermediate into a selenol together with the release of glutathione disulfide (GSSG).

**Figure 5 biomedicines-10-00891-f005:**
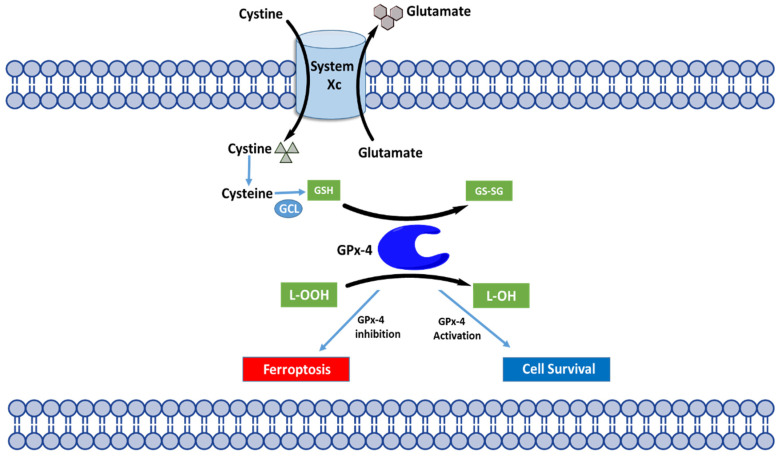
The ferroptotic induced peroxidation of phospholipids is mediated by the cyst(e)ine/GSH/GPX4 regulatory pathway. The system Xc antiporter mediates the exchange of extracellular cystine with intracellular glutamate. When cystine is internalized, it gets reduced to cysteine which is one of the precursors for the biosynthesis of GSH. GPX4 utilizes GSH as a cofactor for the reduction of toxic lipid peroxides into the respective alcohols. Depletion of either GSH or GPX4 will cause an increase in lipid peroxides that will damage the cell membrane, and lead to ferroptotic cell death—a form of cell death implicated in a wide variety of human diseases.

**Table 1 biomedicines-10-00891-t001:** Biochemical Features and Biological Relevance of GPX family.

Mammalian GPX Type	Tissue Distribution	Cellular Localization	Primary Function	Biological Relevance/References
GPX-1	Most abundant and ubiquitously expressed GPx. Highly distributed in the lungs, kidney, red blood cells, and liver.	Cytosol and mitochondria.	Reduces hydrogen peroxides in the cytoplasm at the expense of GSH.	Dampens phosphorylation of phosphatases [13], modulator of the insulin signaling pathway [14], acts in an antiapoptotic manner which can support tumor cell survival [15]
GPX-2	Gastrointestinal tract, endothelial cells (particularly malignant tissues and pluripotent stem cells).	Cytosol	Reduces hydrogen peroxide.	Inhibits inflammation-induced carcinogenesis in the gut [13], but also promotes the growth of some cancers including bladder cancer [16,17,18]
GPX-3	Kidney, lung, heart, muscle.	Plasma	Reduces hydrogen peroxide using GSH, Trx, or Grx.	Deficiency facilitates platelet aggregation and is a risk factor for stroke [13]. Acts as a tumor suppressor in many cancers including lung cancer [19,20]
**GPX-4**	Widespread. Especially testis and spermatozoa kidney, followed by the liver, spleen, pancreas, heart, and brain.	Cytosol, Mitochondria, Plasma.	Reduces hydroperoxides from phospholipids and cholesterol.	Key regulator of ferroptosis [21,22]. Deficiency facilitates male infertility [23,24], Modulator of a rare genetic disorder called SSMD [25]. Implicated in several cancers including CCC and TNBC [26]
GPX-5	Testis, spermatozoa, liver, kidney.	Epididymis	Protects the membranes of spermatozoa from lipid peroxidation.	Deficiency, together with GPX4, decreases male fertility [27]
GPX-6	Embryos and adult olfactory epithelium.	n.d.	n.d.	Reduces the motor defects found in Huntington’s disease [28]
GPX-7	Endoplasmic reticulum	n.d.	Mild glutathione peroxidase activity. Senses ROS levels and transmits redox signals to other thiols.	Contributes to oxidative protein folding in the ER. [29,30]
GPX-8	Endoplasmic reticulum	n.d.	Mild glutathione peroxidase activity. Prevents endoplasmic reticulum oxidation and stress.	Contributes to oxidative protein folding in the ER. [29,30]

Abbreviations: CCC: Clear-cell carcinoma, TNBC: Triple Negative Breast Cancer, Trx: thioredoxin, Grx: glutaredoxin; GSH: glutathione, ER: endoplasmic reticulum; n.d.: not discovered, SSMD: Sedaghatian-type Spondylometaphyseal Dysplasia, p53: tumor-suppressor protein. Bold font highlights Glutathione Peroxidase 4 (GPX4).

**Table 2 biomedicines-10-00891-t002:** Structural Features of Glutathione Peroxidase.

Mammalian GPX Type	Peroxidic Residue	Uniprot Molecular Weight (kDa)	Structure Type	Human Wild-Type Crystal Structure (PDB Code)	Human Mutant Crystal Structure (PDB Code)	Reference (Uniprot Code)
GPX-1	Selenocysteine	22	Homotetramer	n.d.	U46G (2F8A)	P07203
GPX-2	Selenocysteine	21.9	Homotetramer	n.d.	U46C (2HE3)	P18283
GPX-3	Selenocystine	22.5	Homotetramer	n.d.	U46G (2R37)	P22352
**GPX-4**	Selenocysteine	22	Monomer	6ElW	Many mutants (e.g., 7L81, 6HN3, 7L8K, etc.)	P36969
GPX-5	Cysteine	25.2	Homotetramer	213Y	n.d.	O75715
GPX-6	Selenocysteine in Humans. Cysteine in rodents	24.9	Homotetramer	n.d.	n.d.	P59796
GPX-7	Cysteine	20.9	Monomer	2P31	n.d.	Q96SL4
GPX-8	Cysteine	23.8	Monomer	3CYN	n.d.	Q8TED1

Comparison of structural features among the Glutathione Peroxidase family. RCSP Protein Data Bank (PDB) codes. Molecular weight values are reported from the Uniprot database (UniProt https://www.uniprot.org) (accessed on 15 January 2022). The molecular weight of a particular crystal structure may differ depending on the method used for crystallization. Abbreviations. kDa, Kilodalton; n.d., not discovered. Bold font highlights Glutathione Peroxidase 4 (GPX4).

**Table 3 biomedicines-10-00891-t003:** Direct GPX4 Modulators.

Compound	Mode of Action	PubChem CID
(1S,3R)-RSL3	Covalently and irreversibly inhibits GPX4. RSL3 is potent but has poor ADME properties [123]	1750826
DP12--DP19	Not well characterized. Exhibits potency and ferroptosis hallmarks [22]	5728915
Altretamine	GPX4 inhibitor [22]	2123 26186195
DPI10 & ML210	Nitroisoxazole moiety generates a nitrile oxide electrophile that may react with GPX4 [117]	15945537
ML162	Shares the same chloroacetamide moiety as RSL3 but is otherwise very structurally different. Likely to have different off-target effects [22]	3689413
DPI17 & DPI18	Exhibits potency and ferroptosis hallmarks. Likely to be a covalent GPX4 inhibitor [22]	932617
JKE-1674, JKE-1716 & BSC144988	Identical function as DPI10. Nitroisozazole moiety leads to a nitrile oxide electrophilic reaction with GPX4 [117]	145865941
Withaferin A	Acts as a GPX4 inhibitor likely through its electrophilic groups [117]	265237

Established direct modulators of GPX4.

**Table 4 biomedicines-10-00891-t004:** Indirect GPX4 Modulators.

Compound	Possible Mode of Action	PubChem CID
Erastin	Directly inhibits system Xc causing depletion of intracellular GSH, which normally works alongside GPX4 to suppress phospholipid hydroperoxide accumulation [120]	11214940
Erastin Derivatives (Piperazine & Imidazole Ketone Erastin)	Same proposed mode of action as Erasin. These derivatives have improved ADME properties [22]	72710858 & 91824786
RSL5	Displays similar effects as Erastin and may have identical mechanisms, but this has not been experimentally verified [123]	2863472
Sulfasalazine (FDA-approved drug)	Inhibits system Xc, which causes GSH depletion. Low potency and metabolically unstable in vivo [117]	5339
Glutamate	Inhibits system Xc likely by inhibiting one of its kinase targets. May induce necrotic cell death at high concentrations [99]	23672308
Diaryl-isoxazole	Non-competitive System Xc-inhibitor [117,124]	n.a
Engineered human cyst(e)inase	Systemic Depletion of Cysteine [117,124]	n.a
Tac-beclin1	System Xc-inhibitor [117,124]	n.a
Lanperisone (FDA-approved drug)	Inhibits cystine uptake, Causes GSH depletion [117,124]	198707
Sorafenib	Inhibits system Xc, Causes GSH depletion. It also activates NRF2 against ferroptosis [22,121]	216239
FINO2 and FIN56	Does not directly target GPX4, system Xc, or CoQ10. Rather, it oxidizes iron which leads to the subsequent inactivation of GPX4 activity [124]	n.a

Established indirect modulators of GPX4. Abbreviations. NRFT: The nuclear factor erythroid 2-related factor 2.

## Data Availability

Data are contained within the article.

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
