# Peer review of "The Selenoprotein Glutathione Peroxidase 4: From Molecular Mechanisms to Novel Therapeutic Opportunities"

_biomedicines, 2022, doi:10.3390/biomedicines10040891_

Round 1
Reviewer 1 Report
The manuscript submitted by Weaver and Skouta entitled “The Selenoprotein Glutathione Peroxidase 4: From Molecular 2 Mechanisms to Novel Therapeutic Opportunities” reviews the biochemical properties of the GPX4 and its therapeutic potential. They review this by showing the basic biological features of GPX and its biochemistry and molecular biology. Most importantly, GPX4 is a chief regulator of ferroptosis which plays vital role in lipid peroxidation. The authors had a clear review. The manuscript is well written. The authors have finally concluded with the possible therapeutic use of this GPX4.
Author Response
Reviewer #1.
Comments and Suggestions for Authors
The manuscript submitted by Weaver and Skouta entitled “The Selenoprotein Glutathione Peroxidase 4: From Molecular 2 Mechanisms to Novel Therapeutic Opportunities” reviews the biochemical properties of the GPX4 and its therapeutic potential. They review this by showing the basic biological features of GPX and its biochemistry and molecular biology. Most importantly, GPX4 is a chief regulator of ferroptosis which plays vital role in lipid peroxidation. The authors had a clear review. The manuscript is well written. The authors have finally concluded with the possible therapeutic use of this GPX4.
Author response. Thank you for your positive and encouraging comments. I hope the broad scientific community of Pharmaceuticals Journal will value this work as well.
Reviewer 2 Report
This review was well written, however, there are several issues need tobe addressed. A revision is suggested.
- Glutathione peroxidase 4 has a major role in protecting mitochondria, please discuss this issue.
- Please discuss clinical implication of this review.
- Please strength the role of Glutathione peroxidase 4 in human diseases, please focus on oxidative injuries
- Please discuss how Glutathione peroxidase 4 regulates oxidative stress or antioxidant enzymes. If oxidative stress or ROS regulate back Glutathione peroxidase 4 expression.
- Please discuss the relation between Glutathione Peroxidase 4, Ferroptosis and mitochondria.
Author Response
Reviwer#2
Comments and Suggestions for Authors
This review was well written, however, there are several issues need to be addressed. A revision is suggested.
Author response. Thank you for your positive comments regarding the writing style. Below, you will find our revisions as requested.
Comment #1. Glutathione peroxidase 4 has a major role in protecting mitochondria, please discuss this issue.
Comment #5. Please discuss the relation between Glutathione Peroxidase 4, Ferroptosis and mitochondria.
Author response. Thank you for the great suggestion which deeply helped add additional useful information that was missing, alongside some fitting references. The following new section entitled “GPX4, ferroptosis and mitochondria ’’ was included under section 4.5.
Therefore, we added the following section:
4.5. GPX4, ferroptosis, and mitochondria.
The mitochondria is a double-membrane organelle that plays a vital role in energy metabolism . As discussed in greater detail in section 2.6 of this review, the energy-producing cellular processes such as the mitochondrial tricarboxylic acid cycle (TCA) cycle and the electron transport chain (ETC) create ROS that has the potential to be toxic to cells if not regulated properly by antioxidant defense mechanisms such as GPX4 [125,126]. Due to this relationship with ROS, the mitochondria may also be thought of as a key component of cell death pathways including ferroptosis. The appearance of shrunken mitochondria together with an increased membrane density are distinctive hallmarks of ferroptosis recognized since the first discovery of ferroptosis [100]. A comprehensive analysis of the mitochondrial metabolic processes that may facilitate ferroptosis, in addition to the mitochondrial defense systems (most notably GPX4) was recently conducted [127]. In summary, the mitochondria may be related to ferroptosis exclusively due to its metabolic processes that facilitate ferroptosis. However, the mitochondria may not be required for ferroptosis as in mitochondria-depleted cells, toxic accumulation of lipid peroxides are sufficient in inducing ferroptosis [127]. Furthermore, the mitochondria play a complex role in ferroptosis, and the details remain unknown. The mitochondria may be involved in the cysteine-depletion induced aspect of ferroptosis, rather than the GPX4 inactivation-induced pathway of ferroptosis. This hypothesis is a result of the discovery that GPX4 inactivation-induced ferroptosis is caused mostly by nonmitochondrial lipid peroxidation [63].
Comment #2. Please discuss clinical implication of this review.
Author response. Thank you for the comment. The clinical implication of this review was addressed throughout the entire manuscript (e.g., table 1 represents the GPX4 clinical implication in rare genetic disorders called Sedaghatian-type Spondylometaphyseal Dysplasia (SSMD) SSMD [see ref 59]).
Table1: we replaced “Biological Relevance” By “clinical implication”
Ref 59: clinical relevance of GPX4
- Liu, H.; Forouhar, F.; Seibt, T.; Saneto, R.; Wigby, K.; Friedman, J.; Xia, X.; Shchepinov, M.S.; Ramesh, S.K.; Conrad, M.; et al. Characterization of a Patient-Derived Variant of GPX4 for Precision Therapy. Nat. Chem. Biol. 2022, 18, 91–100, doi:10.1038/s41589-021-00915-2.
To further address the reviewer comments, we added a few sentences in section 5. Conclusions and Perspectives
In section 5. Conclusions and Perspectives, we added the following:
Lines (555- 556): Ferroptosis relevance span a multitude of disorders ranging from cancers to neurodegenerative disorders, thereby suggesting the role of GPX4 in these disorders as well.
Lines (555- 556): Notably, very recent studies have investigated GPX4 as a target for precision therapy in a rare genetic disorder. This rare neonatal lethal disorder Sedaghatian-type spondylometaphyseal dysplasia (SSMD) has been linked to four GPX4 variants [59]. Although the mechanism of pathogenesis is unclear, it is thought that the clinical phenotypes of SSMD patients involve complete loss of GPX4 enzymatic function.
Comment #3. Please strength the role of Glutathione peroxidase 4 in human diseases, please focus on oxidative injuries.
Comment #4. Please discuss how Glutathione peroxidase 4 regulates oxidative stress or antioxidant enzymes. If oxidative stress or ROS regulate back Glutathione peroxidase 4 expression.
Author response. Thank you for the suggestion. To address comments # 3 and 4, we added additional information regarding oxidative injuries and how gpx4 regulates oxidative stress under the existing section 2.6. Role of Oxidative Stress in Biology (Line 161).
Therefore, we added the following section
Lines (166-1667):
GPX’s are well-known for their role in disorders characterized by oxidative injuries [1,8,13]. Recent studies showed that oxidative injuries were involved in neurodegeneration such as…..
Lines (173-195):
The imbalanced ROS in the body can cause damage to DNA, lipids, and protein, all of which would cause injury to the cell and can even lead to cell death. The main cause of oxidative stress is an increased production or accumulation of ROS and RNS, a known class of highly reactive compounds containing oxygen and nitrogen respectively. This ROS class of compounds contains radicals and non-radical species including superoxide hydroxyl radicals and hydrogen peroxide, respectively. ROS are mainly derived from the mitochondria where they are regularly produced as a natural byproduct of metabolism [38]. During energy metabolism, the mitochondria’s central role is in oxidative phosphorylation which generates ATP through processes including the TCA cycle and the electron transport chain (ETC). A recent paper made the discovery that inhibition of the mitochondrial TCA cycle and ETC minimized lipid peroxide accumulation and ferroptosis. [125]. Due to the large abundance of ROS in the mitochondria, antioxidants such as GPX4 plays a major role in protecting the mitochondria from oxidative damage. Indeed, GPX4 knockdown lead to increased mitochondrial ROS levels and a subsequent decrease in mitochondrial membrane potential [126]. Interestingly, ROS play a dual role in the human body. An excess of ROS may cause the oxidation of macromolecules and organelles in the body, leading to oxidative stress.
However, ROS may also have non-deleterious effects. They often function as physiological regulators of intracellular signaling pathways, mediate the redox modifications of proteins, and function as intracellular messengers, all crucial roles in the human body. Since ROS is both important in human physiology and can elicit negative effects, the proper regulation of ROS/redox homeostasis is crucial. This maintenance of cellular homeostasis is achieved primarily through the glutathione peroxidase and other oxidoreductase families, among dietary antioxidants and tumor suppressors. Therefore, glutathione peroxidases play a crucial role in health and diseases.
We also added new references:
(Line 880-884)
- Gao, M.; Yi, J.; Zhu, J.; Minikes, A.M.; Monian, P.; Thompson, C.B.; Jiang, X. Role of Mitochondria in Ferroptosis. Mol. Cell 2019, 73, 354-363.e3, doi:10.1016/j.molcel.2018.10.042.
- Cole-Ezea, P.; Swan, D.; Shanley, D.; Hesketh, J. Glutathione Peroxidase 4 Has a Major Role in Protecting Mitochondria from Oxidative Damage and Maintaining Oxidative Phosphorylation Complexes in Gut Epithelial Cells. Free Radic. Biol. Med. 2012, 53, 488–497, doi:10.1016/j.freeradbiomed.2012.05.029.
Finally, as requested by the editor, we rewrite all sections highlighted in yellow.

Round 2
Reviewer 2 Report
my questions had been well addressed this submission is acceptable